# Resveratrol Ameliorates Testicular Histopathology of Mice Exposed to Restraint Stress

**DOI:** 10.3390/ani9100743

**Published:** 2019-09-29

**Authors:** Sheeraz Mustafa, Quanwei Wei, Wael Ennab, Zengpeng Lv, Korejo Nazar, Farman Ali Siyal, Saif Rodeni, Ngekure M. X. Kavita, Fangxiong Shi

**Affiliations:** 1College of Animal Science and Technology, Nanjing Agricultural University, Nanjing 210095, China; sheerazmustafa786@gmail.com (S.M.); weiquanwei@njau.edu.cn (Q.W.); 2016105109@njau.edu.cn (W.E.); lvzengpeng@njau.edu.cn (Z.L.); siaf.luawmsian@gmai.com (S.R.); m15695217055@163.com (N.M.X.K.); 2Faculty of Animal Husbandry and Veterinary Science, Sindh Agriculture University Tando Jam, Tando Jam 70060, Pakistan; nakorejo@sau.edu.pk (K.N.); drfarmansial@gmail.com (F.A.S.)

**Keywords:** restraint stress, histopathology, resveratrol, testis

## Abstract

**Simple Summary:**

The search for effective medicines is challenging. Resveratrol is a phytoalexin, and its function remains unelucidated. Therefore, we undertook the present study to investigate reproductive disturbances due to restraint stress in mice and whether resveratrol plays an anti-stress role. Our results confirmed that resveratrol plays a potential role in the reduction of stress in mice.

**Abstract:**

We evaluated immobilization stress and resveratrol supplementation in immature male mice at 30 days of age for 15 consecutive days. Fifty Swiss mice were divided into five groups (10 mice each): Controls, restraint stress (RS), restraint stress + vehicle (RS + V), RS + 2 mg/kg, and RS + 20 mg/kg. We determined results on the basis of hematoxylin and eosin (H&E), “Periodic acid-Schiff” staining, and TUNEL assay. The results indicated that immobilization stress significantly decreased body weight, testis weight, and water/food intake compared to the control; while resveratrol ameliorated these effects. The quantitative histologic evaluation of the seminiferous tubule diameter, luminal diameter, area of seminiferous tubules, area of tubule lumen, epithelial height, Leydig cell number, and the width of the tunica albuginea were similarly decreased after exposure to RS. These parameters recovered back to normal in the RS + 2 mg/kg group. The development of spermatogenesis was significantly delayed in the RS, RS + V, and RS + 20 mg groups based upon our evaluation score system. However, we observed no significant differences in the RS + 2 mg group compared with the control group. The number of TUNEL-positive cells also significantly decreased in the RS + 2 mg/kg group. In conclusion, we found that the administration of 2 mg/kg was an effective dose against immobilization stress in mice.

## 1. Introduction

The process of unconscious and physiologic deterioration is one of the most common mental disorders and is characterized by affective distress [1]. Approximately 20% of the human population suffers from distress [2]. Therefore, the relation between restraint (immobilization) stress (RS) and sexual behavior in male rats has been studied before [3]. Immobilization stress is a combination of both physical and mental stress, where movement is constrained and the individual is separated from all groups in a confined area [4]. In previous studies, researchers suggested that separation inhibits the hypothalamic-pituitary-adrenal axis and the hypothalamic-pituitary-testicular axis; and that the male reproductive capacity is restrained by restraint stress [5].

Furthermore, the number of germ cells in stage VII of spermatogenesis, testicular weight, concentrations of ascorbic acid and tocopherol, increased counts of abnormal sperm, and the degeneration of seminiferous tubules were significantly dependent on restraint stress [6]. In addition, restraint stress was used as a method to uncover stress effects on reproduction in rodents; and several types of studies showed that the weight of the testes decreased and negatively affected the progressive epididymal sperm and elevated adrenal weight due to stress in rats [7]. Likewise, decreases in the production of sperm, inflation in the ratio of morphologically abnormal sperm, reduced levels of serum testosterone and luteinizing hormone levels, stage-specific apoptosis of germ cells, and the expansion of cytoplasmic vacuolization in Leydig cells were shown to be due to stress in animal models [8]. Due to the characteristics of the tunica albuginea, a fairly strong and inelastic layer, the testis is a conceptually threatened organ; and in some studies, ischemic–reperfusion-related tissue changes can lead to edema with reduced perfusion pressure, which can perpetuate the already compromised blood supply to the testes [3]. However, investigators have found that the tunica albuginea of the human testis contains sufficient contractile elements and demonstrates fundamental regional differences in the contractility [9]. Studies of pharmacologic nutritional substances produced by living organisms have opened a new era of research.

With early observations which show that the histologic parameters of testes can be improved by using hydrophilic or lipophilic antioxidant molecules, a great deal of attention has been paid to studying their impact on male reproduction in diseased or healthy states. Resveratrol (3, 4, 5 trihydroxystilben) is a high-concentration phytoalexin that occurs naturally; and is found in berries, nuts, and medicinal plants; and in grape skin and red wine, in particular [8]. In addition, supplementation with resveratrol can cause an erection of the penis and increase levels of blood testosterone, sperm count, and epididymal sperm motility [10]. Additionally, it has been shown that spermatogenesis is enhanced by stimulating the hypothalamic-pituitary-gonadal axis with resveratrol, with no unfavorable effects [11]. Such resveratrol properties are associated with the inhibition of reproduction in connection with cell cycle inhibition and apoptotic cell death, typically detected in vitro at concentrations ranging from 10 to 300 μm [12,13]. In the present study, the purpose was to investigate histologically the role of resveratrol against potential adverse effects of chronic restraint stress on male mouse testis. Resveratrol may thus play a role in the field of reproduction to reduce the chances of infertility in males.

## 2. Materials and Methods

The experiment was completed under controlled conditions at the Nanjing Agricultural University animal house, including 7 days of quarantine period before the start of the experiment, 15 days of restraint stress, histologic measurements, and data analysis.

### 2.1. Animals and Experimental Design

Thirty-day-old male Swiss ICR mice were purchased from the Qinglongshan Laboratory Animal Company (Nanjing, China). The experimental protocols for mice were in accordance with the Guide for the Care and Use of Laboratory Animals organized by the Nanjing Agricultural University Authorization Committee for Institutional Animal Care and Use. Approval Numbers: 31572403 and 31402075).

Experimental mice were divided into 5 groups of 10 in each group. Young adult Swiss ICR male (30–35 g) were reared under managed conditions that included a 12-h light-dark cycle, 60–70% humidity, and room temperature at 22–23 °C. Suitably balanced rodent feed pellets were given to all mice, and drinking water was provided ad libitum. Animals were acclimated for at least 7 days before the beginning of the experiment. According to previously reported methods, mice were physically restricted in 50-mL conical centrifugal tubes (0.4 cm diameter, various holes drilled for ventilation), [14]. In the absence of food and water, all stressed mice were confined to the tubes for 5 h per day on consecutive days for 15 days [15]. Groups were designed as follows:
(Group 1) Control group—mice without exposure to restraint stress (RS) and treatment. (Group 2) RS group—mice were confined in a conical tube for 5 h per day. (Group 3) RS + V—mice were confined to a conical tube for 5 h per day and 10 μL of vehicle (V) was provided to each mouse by gavage.(Group 4) RS + 2 mg—mice were confined in a conical tube for 5 h per day and 10 µL of resveratrol (2 mg/kg) was given by gavage to each mouse.(Group 5) RS + 20 mg—mice were confined in a conical tube for 5 h per day and 10 µL of resveratrol (20 mg/kg) was given by gavage to each mouse.

All animals were sacrificed after 15 days of stressful immobilization. Four percent paraformaldehyde was used to fix all samples for histologic analysis after the collection and weighing of testes.

### 2.2. Drugs

Resveratrol (Sigma Chemical Co., St Louis, MO, USA) was diluted to 5% in vehicle (CMC-sodium carboxymethyl cellulose) and administered by oral gavage for 15 days, consecutively [16]. The Beijing Institute for Pharmacology and Toxicology (China) supplied sodium carboxymethyl cellulose with *trans*-resveratrol for oral gavage at 0.5% in CMC.

### 2.3. Measurements for Indices of Feed and Water

For each group, the feed intake (g) and the water intake (mL) was described for 15 days consequently [15,17].

### 2.4. Body and Testis Weights

Body weights and testes weights were measured to assess restraint stress effects.

### 2.5. Seminiferous Tubule Diameters and Areas, Tubular Lumen, and Seminiferous Epithelium Heights

Histologic analysis was performed according to previous studies [15,18]. Dehydration of the tissues was done by a graded series of ethanol and xylene and then cleared and infiltrated by paraffin wax. The tissue sections were cut at 5-μm thickness and stained with hematoxylin and eosin (H&E) after 24–48 h in 4% paraformaldehyde. Changes in histopathology were observed at magnifications of 200× and 1000× under an oil immersion microscope (Nikon ECLIPS 80i and OLYMPUS BX51 Tokyo, Japan). For each animal, 25 cross-sections of the most circular seminiferous tubules were photographed (20× objective lens); and in each section, the diameter and radius were measured. In addition, the mean value of 2 seminiferous epithelium heights was obtained by measuring their orthogonal positions. For Leydig cells, 10 sections per sample were examined by using Image j analysis software and a 100× objective lens.

### 2.6. TUNEL Staining

Apoptosis is the end result of injury to the testis. However, the initial mechanism inducing apoptosis is not always definitive. Cellular apoptosis was determined using the terminal deoxynucleotidyl transferase-mediated dUTP nick-end labeling (TUNEL) assay. The apoptotic index was calculated as the percentage of cells that tested TUNEL-positive [19].

### 2.7. Histologic Assessment of Seminiferous Tubules and Maturation

To evaluate seminiferous tubules for spermatogenesis, we used the Johnsen scoring system [20]. Twenty seminiferous tubules were examined in each cross-section and a score of 0–5 was assigned and described.

### 2.8. Statistical Analysis

For the analysis of statistics, Graph Pad Prism 7 was used. All data are presented as mean +/− standard deviation. We measured all areas and diameters on the basis of geometric constant “Pi” square root (A = π √^2^) (A = 3.14 √^2^). In order to compare histopathologic parameters and restraint stress values, we performed a 1-way ANOVA followed by Tukey’s multiple-comparison and a Bonferroni’s multiple comparison post hoc test was performed to determine individual differences. The significance level for our statistics was *p* < 0.05.

## 3. Results

### 3.1. Comparison of Anthropometric Parameters

Difference in the food and water intake was observed in RS, RS + V, and RS + 20 mg/kg groups as compared to the control during the experimental period (Figure 1A,B). Testicular weight in the RS + V group was significantly decreased in comparison to the control, RS + 2 mg/kg, and RS + 20 mg/kg groups. The body weights of the mice in RS, RS + V, and RS + 20 mg/kg groups were significantly different from the control group, as described in Figure 2A,B.

### 3.2. Quantitative Histologic Evaluations

The tubular diameter was significantly increased in the seminiferous tubules of the RS group and slightly increased in the RS + V and RS + 20 mg/kg groups relative to the control group; while the RS + 2 mg/kg group showed a non-significant difference from the control group. The tubular area of the seminiferous tubule sections of each group showed an extensively increased area of tubules in the RS and RS + V groups. Although the RS + 2 mg/kg group area was similar to the control group, that of the RS + 20 mg/kg group was increased. The epithelial height of the seminiferous tubules in the groups of RS and RS + V mice was significantly different from the controls. There was no difference in the groups of RS + 2 mg/kg and RS + 20 mg/kg mice. The luminal diameters of RS and RS + V and RS+20 mg/kg groups were slightly increased as compared to control and RS + 2 mg/kg groups. The tubular luminal area was significantly increased in the RS group, slightly increased in RS + V mice, and non-significant in RS + 2 mg/kg and RS + 20 mg/kg groups. The Leydig cell area was significantly increased in the RS and RS + V groups in comparison to the control group; while this index recovered in the groups of RS + 2 mg/kg and RS + 20 mg/kg mice. The width of the tunica albuginea in RS, RS + V, and RS + 20 mg/kg mouse groups was significantly increased, while in the RS + 2 mg/kg group it was not different from the control group. All data are shown in Figure 3, Table 1. 

### 3.3. TUNEL Assay

The testicular tissue for TUNEL analysis showed that apoptosis germ cells were confined to stage I–III, IV, and V, including the Leydig cells. The number of TUNEL-positive cells in the testes markedly increased in RS, RS + V, and RS + 20 mg groups (Figure 4). TUNEL positive Leydig cells were detected in the sections of RS, RS + V, RS + 2 mg/kg and RS+20 mg groups. Apoptotic cells were not detected in spermatids located close to the lumen. The numbers of TUNEL-positive spermatogonia per seminiferous tubule of the RS, RS + V, and RS + 20 mg/kg groups increased relative to those of the control groups (*p* < 0.05, Table 2, Figure 4). However, the RS + 2 mg/kg group showed a reduction in the apoptotic cells when compared with RS, RS + V, and RS + 20 mg/kg groups (*p* < 0.05) and when compared to the control group testis presented in Table 2, Figure 4. 

### 3.4. Seminiferous Tubule Scores 

The group of control animals exhibited more in the number of spermatogonia, primary and secondary spermatocytes, round sperm, elongated sperm, and spermatozoa than the RS and RS + V groups. The average score for the seminiferous tubules in the RS + 2 mg/kg group was significantly greater than the RS and RS + V and RS + 20 mg/kg groups as shown in Figure 5, Table 3 and Table 4.

## 4. Discussion

Immobilization stress is one of the principal stress-research models used in mice. At present, the prevalence of apoptosis within germ cells of the testis is unelucidated. Results showed that restraint stress was associated with changes in body and testicular weights and that resveratrol could effectively reverse these deficiencies in organ weights after immobilization stress. In addition, we are the first to study the adverse effects of a relatively low dose of resveratrol (2 mg/kg) on male reproductive parameters and mild effects correlated with a higher dose of resveratrol (20 mg/kg) during restraint stress in Swiss mice. Restraint stress is biomedically critical in male reproductive failure, and previous research in our laboratory has also shown that various factors, including sex hormone imbalances, cause pathologic changes in the testis and disorders of the male reproductive system after exposure to stress [15]. All parameters of the stressed group male reproductive system, including body weight in mice, are significantly distinct to those in previous studies [21,22,23,24,25,26]. Many previous studies showed that increased corticosterone mediated restraint-induced weight loss through increasing glycolysis and lipolysis [27,28,29]. In a previous study, immobilization stress rapidly increased the activity of the hypothalamus-pituitary-adrenal axis and suppressed sperm motility by disturbing hypothalamus-pituitary-gonadal axis activity [30]. However, the mechanisms underlying these phenomena are presently unclear. 

In the present study, the decreases in some results (i.e., bodyweight, food intake, and fluid intake) might be due to androgenic hormones and/or neurohormones, accelerating corticosterone secretion and immediately influencing the physiology of the mice. These influences may also affect the epididymis or neuronal circuitry through the sperm neuronal receptor in restraint-stressed mice, and result in the reduction of tubular diameter, tubular area, epithelial height, luminal diameter, luminal area, Leydig cell area, and width of the tunica albuginea. Different investigators have also noted that the male reproductive capacity was regularly decreased [31,32] due to chemical and physical stresses, which supports our study [33]. Our study also proved the curative results of lower-dose resveratrol 2 mg/kg and that the notably higher dose of 20 mg/kg did not circumvent the losses in the tubular diameter and area. In the present study, the width of the tunica albuginea in the stressed groups was significantly increased compared to controls and groups treated with the lower dose of resveratrol, while in the group that was administered the higher dose of resveratrol, the result was significantly different from the control group. Previous studies showed that the human tunica albuginea enclosed the testis with adequate contractual elements and that there were significant regional differences in contractility [9]. Additionally, it is known that the weight and size of the testis and spermatogenesis depend upon testosterone [34]. Our results showed that the Leydig cell area dramatically increased in RS and RS + V groups compared to control groups, which also relates to the previous report [35]. Elevated biogenesis of mitochondria is an adaptive response of repeatedly stressed rats, as was the recovery of testosterone-producing Leydig cells in RS + 2 mg/kg treated mice [36].

In our study RS, RS + V, and RS + 20 mg groups showed highly significantly apoptosis, as in previous studies [37], i.e., testicular tissue analysis using TUNEL showed that apoptosis was confined to the basal germ cells, indicating the suppression of spermatogenesis. In the testis, the number of TUNEL-positive cells increased significantly in a stressed group, while the apoptotic cells of the lower dose (2 mg/kg) in our study were fewer in number. This was the first study to use score in restraint stress with lower and higher doses of resveratrol, and we found that the percentage of the calculated scores were 5, 4, 3, 2, 1, and 0 in multiple cross-sections of seminiferous tubules, the groups of RS, RS + V, and RS + 20 mg were significantly not higher in score. 

## 5. Conclusions

Restraint stress exerts negative effects on health, including the testicular structure of mature mice. Plants contain natural elements, minerals, and vitamins that ensure a secure daily requirement and encourage good health when taken at low doses. Oral gavage with a relatively low dose has curative influences on health and male reproductive parameters in stressed situations in comparison to a higher dose; this is because when the same drugs are taken at a large dose, they become toxic and at subcellular levels generate adverse effects on cells. Resveratrol is similarly useful for health, but its health advantage is dependent upon dosage. Low doses of resveratrol protect animals and humans from various diseases, while higher doses may harm health. We suggest further study on the influence of restraint stress on male fertility and also recommend that future studies should be focused on the underlying reasons for the increased numbers of blood neovascularization in the testes due to stress. We believe that our investigations will pave the way for studying anti-stress mechanisms on testicular structure and function.

## Figures and Tables

**Figure 1 animals-09-00743-f001:**
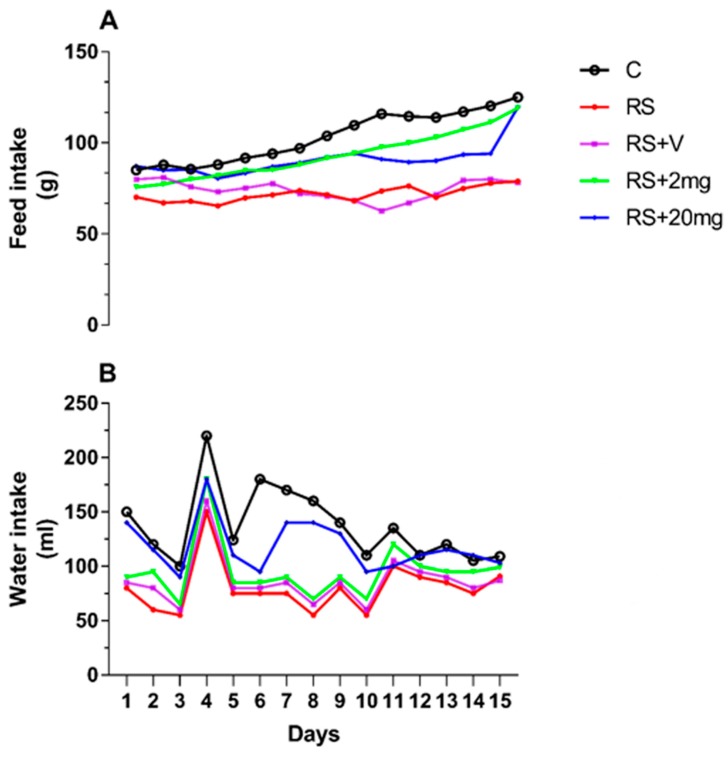
(**A**) Feed and (**B**) water intake of control (C), restraint stress (RS), restraint stress +vehicle (RS + V), RS + 2 mg, and RS + 20 mg groups.

**Figure 2 animals-09-00743-f002:**
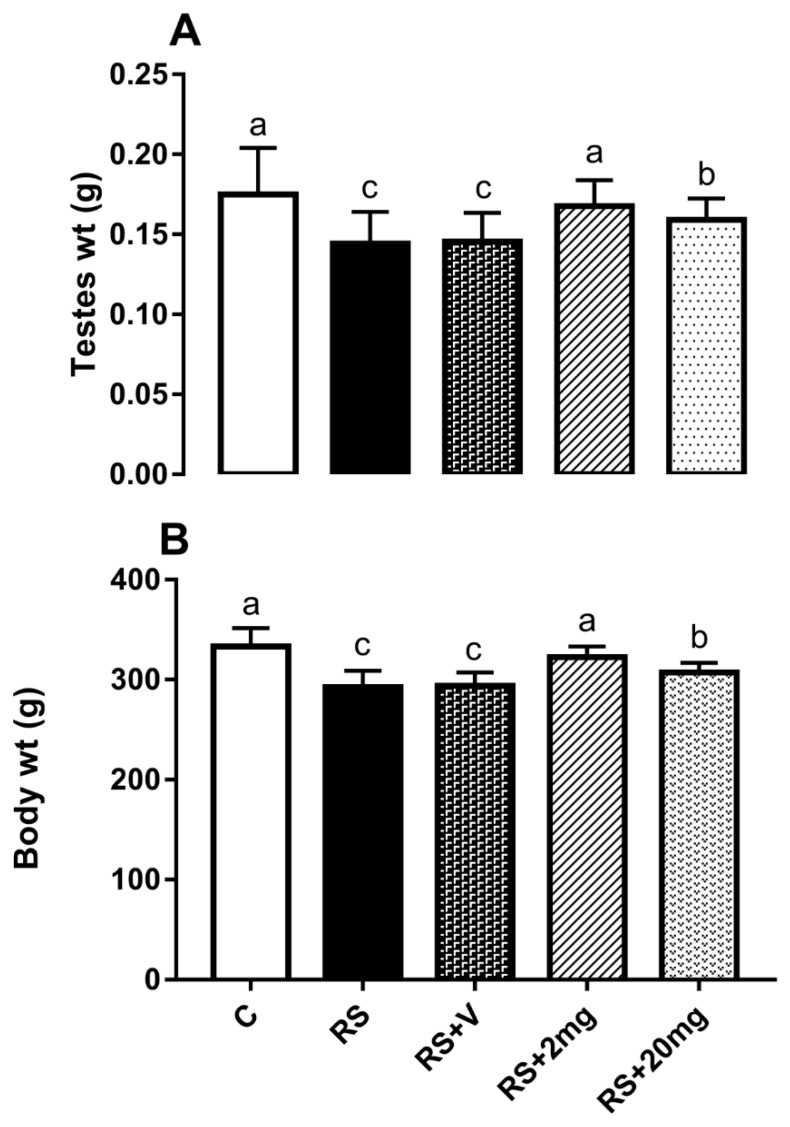
(**A**) Body weight and (**B**) testes weight of control (C), restraint stress (RS), restraint stress + vehicle (RS + V), RS + 2 mg, and RS + 20 mg groups. The statistical differences were determined by one-way ANOVA followed by Tukey’s multiple comparison test. Different superscript letters represent significant differences among groups (*p <* 0.05).

**Figure 3 animals-09-00743-f003:**
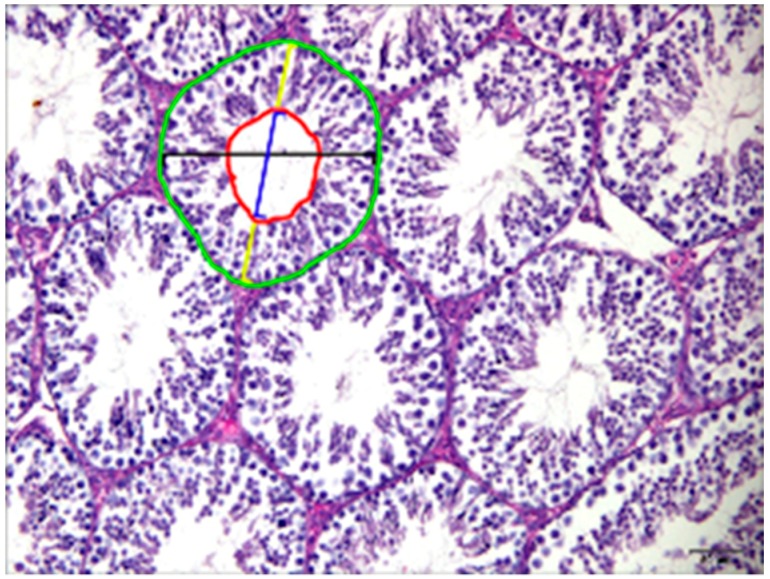
Photomicrograph of mouse testis showing histologic measurements. Area of seminiferous tubules (green circle), diameter of seminiferous tubules (black line), area of the lumen (red circle), diameter of the lumen (blue line), epithelial height (yellow lines).

**Figure 4 animals-09-00743-f004:**
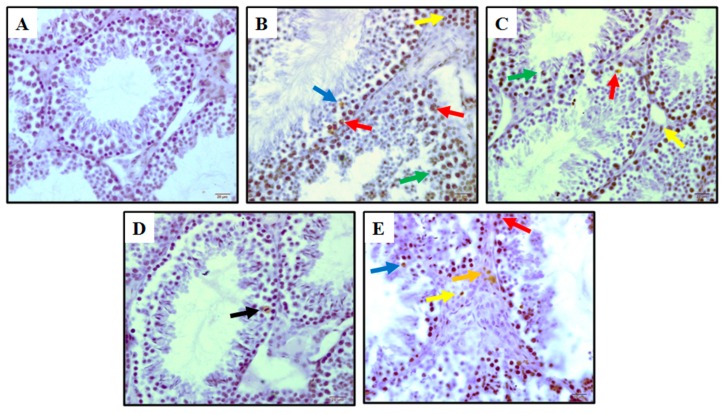
Photomicrographs of histologic sections of mouse testis. (**A**) = control group, (**B**) = RS group, (**C**) = RS + V group, (**D**) = RS + 2 mg group, and (**E**) = RS + 20 mg. TUNEL-positive germ cells were rare in control testes. The blue arrow designates stage IV and the red arrow shows a stage V tubule with spermatogonia (presumably B1), these are the most commonly labeled germ cell types. Large apoptotic pachytene spermatocytes can be clearly identified and are the most common cell type labeled with the TUNEL assay at these stages. The green arrow shows stage VII of residual apoptotic pachytene spermatocytes; the yellow arrow shows apoptotic Leydig cells; the black arrow shows a residual body; the orange-colored arrow shows slightly apoptotic Leydig cells. Scale bar = 20 mm.

**Figure 5 animals-09-00743-f005:**
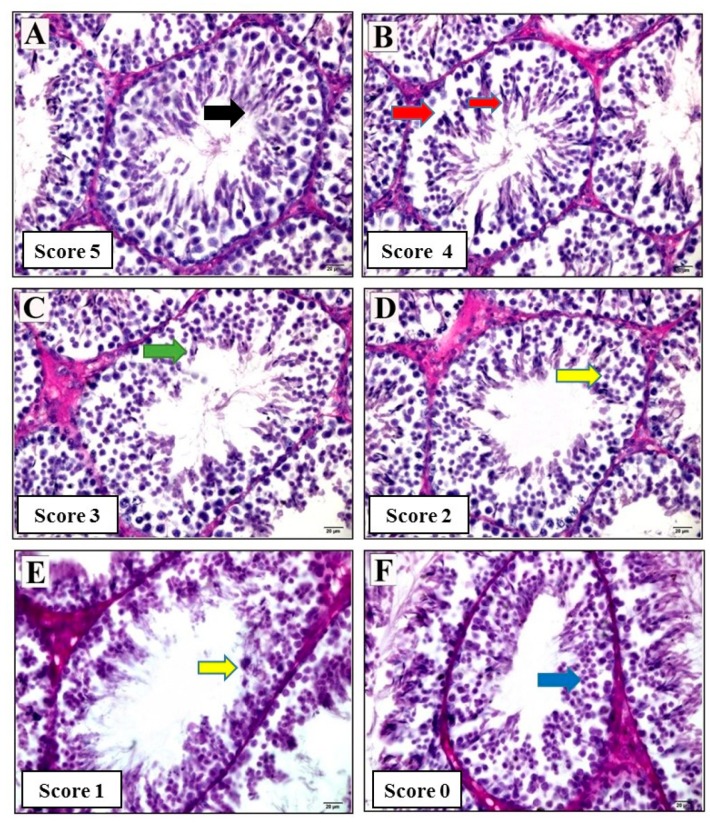
Photomicrographs of model examples used for standardization of scores in seminiferous tubule cross-sections. (**A**) Black arrow = complete spermatogenesis with mature sperm cells. (**B**) Red arrow = disorganized epithelium with some sperm cells (small red arrow). (**C**) Green arrow = presence of few sperm (<5 to 10). (**D**,**E**) Yellow arrow = absence of sperm cell and the presence of spermatids. (**F**) Blue arrow = absence of sperm cells or spermatids and presence of spermatocytes. *p* < 0.05. Tissues were stained with hematoxylin-eosin (H&E). Bar = 20 μm.

**Table 1 animals-09-00743-t001:** Quantitative histologic evaluation of seminiferous tubules in control (C), restraint stress (RS), restraint stress + vehicle (RS + V), restraint stress (RS) + 2 mg/kg, and RS + 20 mg/kg groups.

Groups	Tubular Diameter (μm) *N* = 25	Tubular Area (μm^2^) *N* = 25	Epithelial Height (μm) *N* = 25	LuminalDiameter (μm^2^) *N* = 25	Luminal Area (μm^2^) *N* = 25	Leydig Cell Area (μm^2^) *N* = 50	Width of Tunica Albuginea (μm) *N* = 25
C	186.48 ± 23.93	103.67 ± 8.63.15	55.87 ± 10.22	69.10 ± 16.99	49.26 ± 8.47	108.19 ± 30.60	12.99 ± 1.35
RS	150.67 ± 10.64 *	115.93 ± 10.57 *	35.93 ± 7.88 *	82.53 ± 10.06 *	68.33 ± 8.244 *	149.76 ± 34.38 *	15.31 ± 0.60 *
RS + V	155.35 ± 8.10 *	123.04 ± 7.19 *	40.19 ± 7.40 *	78.08 ± 10.11	56.29 ± 6.70 *	127.99 ± 36.13 *	14.72 ± 2.13 *
RS + 2 mg	171.81 ± 14.10	105.81 ± 12.20	50.87 ± 9.26	64.42 ± 12.71	47.40 ± 11.38	113.73 ± 20.61	13.05 ± 1.09
RS + 20 mg	171.93 ± 13.78 *	110.42 ± 6.05 *	51.33 ± 7.07	77.22 ± 13.13	47.66 ± 9.35	114.59 ± 23.26	12.12 ± 0.56 *

Values appear as mean ± SD. “*N*” indicate number of total observations. * indicates significant differences among groups *p* < 0.05.

**Table 2 animals-09-00743-t002:** Observed apoptotic testicular germ cells per 50 Sertoli cell nuclei, and the Leydig cell apoptotic index and frequencies (%) of stages for seminiferous epithelium from round tubule cross-sections in control, RS, RS+V, RS + 2 mg, and RS + 20 mg groups.

Groups	*N*	Stages			Leydig Cells
		I–III	IV	V	
C	50	5 (10%)	7 (14%)	4 (8%)	0 (0%)
RS	50	39 (78%)	36 (72%)	37 (74%)	25 (50%)
RS + V	50	31 (62%)	29 (58%)	30 (60%)	22 (44%)
RS + 2 mg	50	24 (48%)	23 (46%)	20 (40%)	13 (26%)
RS + 20 mg	50	32 (64%)	37(74%)	31 (62%)	18 (36%)
		*p* < 0.05 *	*p* < 0.05 *	*p* < 0.05 *	*p* < 0.05 *

Values are expressed as observed apoptotic stages of testicular germ cell frequency (%) in each column. “*N*” indicates the total number of intertubular cells, * *p*-values of significant differences among groups for each parameter at *p* < 0.05 of a Bonferroni’s multiple comparison post hoc test was performed to determine individual differences.

**Table 3 animals-09-00743-t003:** Criteria of scores for the evaluation of Swiss mouse spermatogenesis.

Score	Description
5	Complete spermatogenesis with mature sperm cells
4	Some sperm cells, with a disorganized epithelium
3	Presence of few sperm (<5 to 10)
2	Absence of sperm cells, presence of spermatids
1	Absence of sperm cells, presence of a few spermatids
0	Absence of sperm cells or spermatids, presence of spermatocytes

**Table 4 animals-09-00743-t004:** Frequency and percentage of seminiferous tubule score 5, 4, 3, 2, 1, and 0 in multiple cross-sections in control, RS, RS + V, RS + 2 mg, and RS + 20 mg groups.

Groups	Number of Sections	Scores					
		5	4	3	2	1	0
C	20	8 (40%)	6 (30%)	2 (10%)	2 (10%)	1 (5%)	1 (5%)
RS	20	1 (5%)	1 (5%)	2 (10%)	5 (10%)	5 (25%)	6 (30%)
RS + V	20	0 (0%)	1 (5%)	5 (25%)	5 (25%)	3 (15%)	6 (30%)
RS + 2 mg	20	4 (20%)	6 (30%)	4 (29%)	2 (10%)	2 (10%)	2 (10%)
RS + 20 mg	20	2 (10%)	2 (10%)	3 (15%)	4 (29%)	5 (25%)	4 (29%)
		*p* > 0.05	*p* > 0.05	*p* > 0.05	*p* > 0.05	*p* < 0.05	*p* > 0.05

Values are expressed as seminiferous tubule scores and percentage in each column. *p* indicates *p* value of significant differences among groups for each parameter at *p* < 0.05 of a Bonferroni’s multiple comparison post hoc test to determine individual differences.

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
