# Peer review of "Resveratrol Ameliorates Testicular Histopathology of Mice Exposed to Restraint Stress"

_animals, 2019, doi:10.3390/ani9100743_

Round 1
Reviewer 1 Report
Corrections and comments below -
Line 18 – typos – should be “Periodic acid-Schiff”, TUNNEL should be corrected to ‘TUNEL’
Line 81 – states “Animals were treated for at least 7 days before the beginning of the experiment.” What were they treated with? It seems to be a misstatement. It should state – “Animals were acclimated for at …..”.
Line 93 – check grammar error. Should be “was given ….”
Line 121 – states scoring levels of 1-6. However, the tables 4 and 5 classify scores as 0-5.
Line 122 – check statement: do you mean “as described”. Is parentheses needed?
Line 127 – check statement: calculation of percentage itself involves dividing over. So the use of word “divided” by total number of cells should be corrected. It can be phrased as “The apoptotic index was calculated as the percentage of cells that tested TUNEL-positive.”
Line 145 – typo error: it should be “histologic”
Line 146 – typo error: it should be “significantly”
Table 2 – can the formatting of data values be modified to place the SD values such that the reader can easily identify the asterisk which represent significance difference among groups. For example, move the SD values closer and next to the mean.
Line 178 – typo error: should be “TUNEL”
Table 3 – There are multiple types of asterisk listed in the table. Please provide statement in legend description about what does that mean.
Line 251 – Figure 3 and Table 5 should be moved ahead of Table 4 so that it is easy for readers to understand the scoring system used in Table 4. If done, the tables will need to be renumbered and also their references in manuscript will also need to be corrected
Line 293-295 – The sentence “All parameters ………previous studies” needs to be rephrased. The statement is unclear and confusing.
Line 295 – The statement “Many previous studies showed that improved corticosterone…” – replace “improved” with “increased”.
Line 325-327 – The sentence is unclear and also incorrect. And these groups listed in this sentence were not higher in score. Describe this sentence clearly and correctly.
Line 327-329 – It is unclear what are authors trying to state in that sentence. Revise or remove the statement.
Line 332 – grammatical error – should be “Plants contain ….
Line 339 – grammatical error – should be ‘recommend’ (not ‘recommended’)
Author Response
Responses to the Reviewers:
Thank you for your encourage and offer for resubmission of our manuscript after extensive revisions and comments. Our response to your points are as follows.
Reviewer 1.
Line 18 – typos – should be “Periodic acid-Schiff”, TUNNEL should be corrected to ‘TUNEL’Response:
Line 81 – states “Animals were treated for at least 7 days before the beginning of the experiment.” What were they treated with? It seems to be a misstatement. It should state – “Animals were acclimated for at …..”.
Response: Statement has been corrected with the replacement of your suggested suitable word acclimated.
Line 93 – check grammar error. Should be “was given ….”
Response: C
Line 121 – states scoring levels of 1-6. However, the tables 4 and 5 classify scores as 0-5.
Response: Scoring levels of states is settled as 0-5 as mentioned in tables 4 and 5.
Line 122 – check statement: do you mean “as described”. Is parentheses needed?
Response: Parentheses are removed after your valuable suggestions.
Line 127 – check statement: calculation of percentage itself involves dividing over. So the use of word “divided” by total number of cells should be corrected. It can be phrased as “The apoptotic index was calculated as the percentage of cells that tested TUNEL-positive.”
Response: The statement is changed as: The apoptotic index was calculated as the percentage of cells that tested TUNEL-positive.
Line 145 – typo error: it should be “histologic”
Response:
Line 146 – typo error: it should be “significantly”
Response: Corrected as your suggestions.
Table 2 – can the formatting of data values be modified to place the SD values such that the reader can easily identify the asterisk which represent significance difference among groups. For example, move the SD values closer and next to the mean.
Response: Revised as your suggestions.
Line 178 – typo error: should be “TUNEL”
Response: Corrected.
Table 3 – There are multiple types of asterisk listed in the table. Please provide statement in legend description about what does that mean.
Response: Multiple types of asterisks were by mistaken due to wrong calculation formula, corrected now.
Line 251 – Figure 3 and Table 5 should be moved ahead of Table 4 so that it is easy for readers to understand the scoring system used in Table 4. If done, the tables will need to be renumbered and also their references in manuscript will also need to be corrected
Response: Revised as your suggestions Thank you!
Line 293-295 – The sentence “All parameters ………previous studies” needs to be rephrased. The statement is unclear and confusing.
Response: The statement has been written.
Line 295 – The statement “Many previous studies showed that improved corticosterone…” – replace “improved” with “increased”.
Response: corrected.
Line 325-327 – The sentence is unclear and also incorrect. And these groups listed in this sentence were not higher in score. Describe this sentence clearly and correctly.
Response: The sentence is made clear and corrected with some modifications to understand the readers easily.
Line 327-329 – It is unclear what are authors trying to state in that sentence. Revise or remove the statement.
Response: Statement is deleted.
Line 332 – grammatical error – should be “Plants contain ….
Response: corrected.
Line 339 – grammatical error – should be ‘recommend’ (not ‘recommended’)
Response: corrected. Thank you!
Reviewer 2 Report
Although this manuscript provides some interesting scientific results several deficiencies should be addressed before acceptance for publication in the Animals.
All the data in Table 1 seems to be the total data of 10 mice.
If you use a total of 50 mice and put 10 mice in one cage, you cannot say that n = 10 data.
Please describe correctly.
In Table 2, why is the area of the RS + V tube unusually small even though the tube diameter is almost the same as the others?
In Tables 2, 3, and 4, there are N = 25, N = 50, and N = 20. Is this the value examined for one mouse?
Or is it the value examined for 25 mice, 50 mice, 20 mice?
If the data was obtained from one mouse, the reliability is low.
The statistical analysis in Tables 3 and 4 should be performed by Multi-way Analysis of Variance.
Author Response
All the data in Table 1 seems to be the total data of 10 mice.Response:
If you use a total of 50 mice and put 10 mice in one cage, you cannot say that n = 10 data. Please describe correctly.
Response: Yes, we revised our explanation about this issue.
In Table 2, why is the area of the RS+V tube unusually small even though the tube diameter is almost the same as the others?
Response: Values of tubular area for the groups of C, RS, RS+V, and RS+20mg, and Luminal area for all five groups were given directly by mistakenly, which are now presented as (μm2) as other areas are given. Bundle of thanks for suggestions.
In Tables 2, 3, and 4, there are N = 25, N = 50, and N = 20. Is this the value examined for one mouse? Or is it the value examined for 25 mice, 50 mice, and 20 mice? If the data was obtained from one mouse, the reliability is low.
Response: No, this 20 is a typographical error, which is corrected as 25 as others. Measurements of 25 and 50 are random collections from each group of mice.
The statistical analysis in Tables 3 and 4 should be performed by Multi-way Analysis of Variance.
Response: Bonferroni’s multiple comparison post hoc test for Analysis of Variance are included in the tables of 3 and 4 to describe better the results. Thank you!
Round 2
Reviewer 2 Report
I have no additional comments.
This manuscript is a resubmission of an earlier submission. The following is a list of the peer review reports and author responses from that submission.
Round 1
Reviewer 1 Report
There is significant language issue in this manuscript. Please consult professional scientific writer who is also proficient in English.
The table and figures should also be corrected and edited to make it more clear and understandable.
You need to explain under Discussion some of the findings which are unusual and not expected - for example high dose did not show any effect despite significant effect of low dose.